# Nivolumab plus Ipilimumab versus Existing Immunotherapies in Patients with PD-L1-Positive Advanced Non-Small Cell Lung Cancer: A Systematic Review and Network Meta-Analysis

**DOI:** 10.3390/cancers12071905

**Published:** 2020-07-15

**Authors:** Koichi Ando, Yasunari Kishino, Tetsuya Homma, Sojiro Kusumoto, Toshimitsu Yamaoka, Akihiko Tanaka, Tohru Ohmori, Tsukasa Ohnishi, Hironori Sagara

**Affiliations:** 1Division of Respiratory Medicine and Allergology, Department of Medicine, Showa University School of Medicine, 1-5-8 Hatanodai, Shinagawa-ku, Tokyo 142-8666, Japan; ookiyookiy@med.showa-u.ac.jp (Y.K.); tetsuya.homma@med.showa-u.ac.jp (T.H.); k-sojiro@med.showa-u.ac.jp (S.K.); tanakaa@med.showa-u.ac.jp (A.T.); ohmorit@med.showa-u.ac.jp (T.O.); tohnishi@med.showa-u.ac.jp (T.O.); sagarah@med.showa-u.ac.jp (H.S.); 2Advanced Cancer Translational Research Institute (Formerly, Institute of Molecular Oncology), Showa University, 1-5-8 Hatanodai, Shinagawa-ku, Tokyo 142-8555, Japan; yamaoka.t@med.showa-u.ac.jp

**Keywords:** nivolumab, ipilimumab, pembrolizumab, programmed cell death ligand 1, non-small cell lung cancer, progression-free survival, overall survival, network meta-analysis, indirect treatment comparison, systematic review

## Abstract

No head-to-head trials have compared the efficacy and safety of nivolumab (Niv) plus ipilimumab (Ipi) combination therapy (Niv+Ipi) and existing regimens with immunotherapies approved as first-line treatment in patients with programmed cell death ligand 1 (PD-L1)-positive previously untreated advanced non-small cell lung cancer (NSCLC). We conducted a network meta-analysis of four relevant Phase Ⅲ trials to compare the efficacy and safety of Niv+Ipi, pembrolizumab (Pem) plus platinum-based chemotherapy (PBC) (Pem+PBC), Pem, Niv, or PBC using Bayesian analysis. The primary efficacy endpoint was progression-free survival (PFS) in patients with advanced NSCLC with PD-L1 expression ≥1%. The primary safety endpoint was the incidence of Grade 3–5 drug-related adverse events (G3–5AEs). Efficacy and safety were ranked using surface under the cumulative ranking curve (SUCRA). With regard to PFS, Niv+Ipi was inferior to Pem+PBC, and superior to Pem, Niv, or PBC alone. SUCRA ranking showed Pem+PBC had the highest efficacy for PFS, followed by Niv+Ipi, Niv, PBC, and Pem. The safety outcome analysis revealed Niv+Ipi was generally well tolerated compared to existing immunotherapy regimens. These results provide clinical information regarding the efficacy and safety of Niv+Ipi and indicate the possibility of the Niv+Ipi combination as a new therapeutic option in PD-L1-positive advanced NSCLC.

## 1. Introduction

The achievements of molecular oncology and immunology over the past decade have fueled an understanding of the onset and progression of lung cancer [1,2]. As a result, in just the last few years, strategies for treating lung cancer have evolved remarkably, particularly in the fields of molecularly targeted therapy and immunological therapy [1,2,3]. Nonetheless, lung cancer remains the leading cause of cancer death in the world. According to the 2018 cancer statistics, of all cancers, lung cancer accounts for 13%, and the 5-year survival rate of lung cancer is a mere 18% [1,4,5]. Although 16% of patients with lung cancer are diagnosed localized in the lungs and the presumed 5-year survival rate for these cancers is 56%, most of the patients with lung cancers are diagnosed in an advanced state, and the 5-year survival rate of these patients is a mere 5% [5,6,7,8]. In particular, non-small cell lung cancer (NSCLC) accounts for 84% of lung cancers, which calls for a strong need for continuous improvement and modification of its treatment strategy [1,5,6,7,8].

In recent years, several immunological treatment regimens, including immune checkpoint inhibitors (ICI), have been approved for lung cancer and are already in widespread clinical application [9,10]. In particular, immunological treatment greatly contributes to the outcome of NSCLC treatment [2,9,10].

Furthermore, recently, the efficacy of nivolumab (Niv), a fully human monoclonal immunoglobulin G4 antibody to programmed cell death-1 (PD-1), plus ipilimumab (Ipi), a fully humanized IgG1 monoclonal antibody to cytotoxic T lymphocyte antigen-4 (CTLA-4), as a combination therapy (Niv+Ipi) in programmed cell death ligand 1 (PD-L1)-positive previously untreated advanced NSCLC has been reported [11]. Prolonged progression-free survival (PFS) has been shown in comparison to either monotherapy with platinum-based chemotherapy (PBC) or nivolumab (Niv) monotherapy [11]. Therefore, it is expected that Niv+Ipi may be a new treatment option for PD-L1-positive previously untreated advanced NSCLC [12].

However, at present, there have been few reports comparing Niv+Ipi to existing regimens with immunotherapy, and the details of the efficacy and safety profiles of Niv+Ipi have not been sufficiently clarified. Although Niv+Ipi is acceptable for treating NSCLC, currently, there is limited justification for choosing Niv+Ipi over other existing immunotherapy regimens, such as pembrolizumab (Pem), nivolumab (Niv), or Pem plus platinum-based chemotherapy (PBC) (Pem+PBC). Comparison and ranking of Niv+Ipi to existing regimens with immunotherapies will provide oncologists with useful information regarding the selection of therapeutics. 

In general, we should conduct a direct head-to-head randomized controlled trial (RCT) to compare drug efficacy and safety. However, RCT studies are time consuming, expensive, and labor intensive. In the absence of a direct head-to-head RCT, an indirect treatment comparison (ITC) can be performed by setting a common comparison target by adopting a statistical method called a network meta-analysis (NMA), using all the data accumulated so far [13,14,15]. By performing ITC, we can obtain useful information more rapidly than from RCTs [13,15]. The advantages of an NMA are that any pair of two treatments could be compared even if there is no existing direct comparison RCT, and the efficacy and safety of each treatment can be ranked [13,14,15,16,17].

The purpose of this systematic review (registration: UMIN-CTR no. UMIN000039784) was to compare the efficacy and safety of Niv+Ipi in PD-L1-positive advanced NSCLC derived from publicly available RCTs based on existing regimens with immunotherapies, such as Pem, Pem+PBC, and Niv with PBC as a common comparator, and to rank the efficacy and safety of these therapeutic regimens using the statistical Bayesian NMA approach.

## 2. Methods 

### 2.1. Systematic Review

A comprehensive literature search was performed to identify published studies of RCTs of Niv+Ipi or immunotherapies for advanced NSCLC from 1946 to the present. On 13 March 2020, four databases (PubMed (https://pubmed.ncbi.nlm.nih.gov/), CENTRAL (https://www.cochranelibrary.com/), EMBASE (https://dialog.proquest.com/professional/login), and SCOPUS (https://www.scopus.com/home.uri) were searched. The keywords NSCLC, immunotherapies, PD-L1, and their Medical Subject Headings (MeSH) terms were used to build a search strategy. We did not restrict publications to those only in English. We presented the strategy used to search PubMed in Appendix B. Additionally, to avoid the risk of missing any relevant studies that met the inclusion criteria, we reviewed the reference lists of the retrieved studies. An email query was sent to the corresponding author if a database did not provide enough information about the study. We also searched EMBASE, CENTRAL, and SCOPUS using the same search strategy.

We performed a review of the references presented in the articles, combined with a manual search of relevant articles, to identify all relevant studies and to minimize publication bias. The study was performed in accordance with the Preferred Reporting Items for Systematic Review and Meta-analysis (PRISMA) statement [18] and the PRISMA extension statement for an NMA [19]. Two investigators (K.A. and T.Y.) independently performed the literature searches. Inclusion and exclusion criteria were adapted by using the PICOS approach for the studies retrieved in this systematic review to address the clinical or methodological heterogeneity between studies and to ensure the validity of the indirect comparison analysis.

### 2.2. Quality Assessment

We adapted the risk-of-bias tool recommended by the Cochrane Collaboration to evaluate the quality of the RCTs included in the analysis [20]. The following listed parameters were evaluated as being high, unclear, or low: (1) random sequence generation; (2) allocation concealment; (3) blinding of participants and personnel; (4) blinding of outcome assessment; (5) incomplete outcome data; (6) selective reporting; and/or (7) other biases.

### 2.3. Inclusion and Exclusion Criteria (Predefined PICOS)

#### 2.3.1. Patients

The objective of this study was to compare the efficacy and safety of Niv+Ipi to existing regimens with immunotherapies in adult PD-L1-positive advanced NSCLC. Therefore, we established the inclusion criteria listed below to faithfully reflect the purpose of this study. To be included in this analysis, the patient group had to meet the following criteria: minimum age of 18 years, histological or cytological confirmation of PD-L1-positive (expression ≥ 1%) previously untreated advanced NSCLC, with an Eastern Cooperative Oncology Group (ECOG) performance status of 0 to 2 (on a 5-point scale, with higher numbers reflecting greater disability). 

#### 2.3.2. Interventions/Comparisons

In order to be included in this study, it was necessary to have at least one of the following as a treatment arm: (1) Pem 200 mg/body weight every three weeks (e3w) plus PBC (cisplatin-based or carboplatin-based chemotherapy); (2) Ipi 1 mg/kg every six weeks; (3) Pem 200 mg/body weight e3w monotherapy; (4) Niv 240 mg/body weight every two weeks monotherapy; and (5) PBC. The above doses have been adopted in the licensed dosage and administration of Phase III trials. A common comparison was assumed to be PBC. This was because it was a standard treatment before the emergence of immunotherapy for NSCLC. 

Atezolizumab, like Niv and Pem, is an ICI with indication for NSCLC. However, in Phase III trials of atezolizumab alone and atezolizumab-containing regimens in NSCLC, no subset analysis was performed in the group with a PD-L1 expression of ≥1%, which was the patient population studied in this NMA. We concluded that incorporating a regimen containing atezolizumab into this NMA was not appropriate in view of the heterogeneity of the target patients. Therefore, immunological regimens containing atezolizumab and atezolizumab monotherapy were not included in the present NMA.

#### 2.3.3. Outcomes

The primary efficacy endpoint for our analysis was PFS, which has been expressed in terms of HR and 95% CrI in patients with PD-L1 expression levels of ≥1%. The secondary efficacy endpoints were PFS in subgroups with PD-L1 expression levels of ≥50%, OS in the patient group with PD-L1 expression levels of ≥1%, and OS in the subgroup with PD-L1 expression levels of ≥50%. The primary safety endpoint was the incidence of drug-related G3–5AEs, which was expressed as the OR and 95% CrI. To rank the efficacy of each treatment, the values of the SUCRA values were calculated for PFS in patients with a PD-L1 expression of ≥1%, and in the subgroup with a PD-L1 expression of ≥50%; for OS in patients with a PD-L1 expression of ≥1%, and in subgroups with a PD-L1 expression of ≥50%; and for the G3–5AEs for all participants in the RCTs, an NMA was also calculated [21]. This analysis was performed not only in patients with a PD-L1 expression of ≥1% but also in the subgroup with a PD-L1 expression of ≥50%. We analyzed these predefined endpoints only if data were available from the included studies. The relevant data were extracted by two authors (K.A. and T.Y,) independently, and, when necessary, a third author (TO) was consulted to resolve any discrepancies.

#### 2.3.4. Study Design

The studies that were considered to be eligible for inclusion in this NMA were defined as randomized, parallel group Phase III studies. At least one predefined efficacy or safety endpoint had to be available in the study for it to be included in the present NMA.

### 2.4. Statistical Method of Network Meta-Analysis

We conducted comparison of Niv+Ipi and the existing immunotherapies for the predefined safety and efficacy endpoints by using the Bayesian NMA method in accordance with established methods formulated by the National Institute for Health and Care [14,15,16]. This statistical method is well-established for NMA and is supported by the National Institute for Health and Clinical Excellence and the Haute Autorité de Santé, as well as by the International Society for Pharmacoeconomics and Outcome Research (ISPOR) guidelines for indirect comparison and NMA [22,23]. The methodology of indirect treatment comparison using the NMA statistical method is useful for comparing treatment regimens in the absence of RCTs performing direct comparisons [24], which has been adapted in a variety of fields of clinical research [25,26,27,28]. For the present analysis, we used the standard method of NMA described by Dias et al. [29,30,31].

The Bayesian model, which assumes inconsistency and heterogeneity between the included studies [30], were adopted for the present NMA. To estimate the posterior distribution of the treatment effects, the Bayesian analysis involved the adaptation of a non-informative prior distribution and Gibbs sampling by using the Markov chain Monte Carlo method. We performed 50,000 iterations, with the first 10,000 considered as burn-in samples. The BGR diagnostic method [32,33] was used to evaluate model convergence. The treatment effect was expressed in terms of HR and OR with 95% CrIs. A 95% CrI was derived from 2.5% and 97.5% of the posterior distribution. If the 95% CrI crossed the invalid line (i.e., the HR or OR was 1), the result was interpreted as statistically insignificant. For each outcome, analysis was conducted in a group of patients with a PD-L1 expression ≥ 1% and in a group of patients with a PD-L1 expression ≥ 50%. These analyses were conducted only if the analysis was possible from the extracted data.

NMA facilitates not only comparison but also ranking of treatment groups. In this NMA, treatments were ranked based on the SUCRA values calculated from a Bayesian analysis [34]. The SUCRA values ranged from 0% to 100%, with higher SUCRA values indicating that the treatment was relatively more effective; a value of 100% indicated that the drug was the most ideal treatment [34]. We performed the analysis using OpenBUGS 1.4.0 (MRC Biostatistics Unit, Cambridge Public Health Research Institute, Cambridge, UK), and STATA (ver. 14, StataCorp, College Station, TX, USA) was utilized for graphical presentation of the results.

### 2.5. Ethical Aspects

We waived institutional review board approval and patient consent because of the nature of the review performed in this study.

## 3. Results

### 3.1. Systematic Review 

We performed a systematic literature review, and identified 4997 articles (506 from PubMed, 1977 from EMBASE, 92 from the Cochrane Central Register of Controlled Trials (CENTRAL), and 2422 from SCOPUS) that satisfied the search criteria; 3487 articles were retained after removal of duplicates. Adoption of the Patients, Intervention, Comparison, Outcome, and Study design (PICOS) approach led to the retention of four studies, one of which compared Niv+Ipi with Niv or chemotherapy (CheckMate-227) [11], another compared Pem and PBC (KEYNOTE-042) [35], and the other two studies compared Pem+PBC and PBC (KEYNOTE-189 and KEYNOTE-407) [21,36]. Figure 1 shows the study selection process, Table 1 shows the key inclusion criteria, and Table 2 shows the primary characteristics of the included studies. 

The common comparative group in Niv+Ipi and studies of existing immunotherapies was PBC. Although study data were available for the NMA of the predefined primary and secondary efficacy endpoints, progression-free survival (PFS) and overall survival (OS), data reported in the subgroup of the threshold of the PD-L1 expression levels, were not sufficient for the NMA of the predefined safety endpoint (G3–5AEs). Hence, the NMA of the safety endpoint was performed only for the overall participants in the included studies of the NMA. We confirmed the preferred model convergence in all the analyses by adopting the Brooks–Gelman–Rubin (BGR) diagnostic method. Figure 2 represents a network map of this NMA.

### 3.2. Primary Efficacy Endpoint: Progression-Free Survival in Patients with a PD-L1 Expression of 1% or More

Niv+Ipi significantly improved PFS compared to Pem, Niv, and PBC, with a hazard ratio (HR) and 95% credible intervals (CrI) of 0.770 (0.632 to 0.930), 0.832 (0.710 to 0.970), and 0.823 (0.708 to 0.950), respectively, whereas Niv+Ipi was inferior in PFS to Pem+PBC with an HR and 95% CrI of 1.784 (1.396 to 2.246). Pem+PBC significantly improved PFS compared to Pem, Niv, and PBC, with an HR and 95% CrI of 0.436 (0.346 to 0.542), 0.474 (0.352 to 0.624), and 0.465 (0.384 to 0.558), respectively. There was no significant difference between Pem and Niv, between Pem and PBC, nor between Niv and PBC, with an HR and 95% CrI of 1.091 (0.845 to 1.387), 1.072 (0.943 to 1.213), and 0.994 (0.798 to 1.225), respectively (Figure 3).

### 3.3. Secondary Efficacy Endpoint: Progression-Free Survival in Patients with a PD-L1 Expression of 50% or More

Niv+Ipi significantly improved PFS compared to PBC, with an HR and 95% CrI of 0.625 (0.488 to 0.788), whereas Niv+Ipi was inferior in PFS to Pem+PBC, with an HR and 95% CrI of 1.617 (1.056 to 2.375). Pem+PBC significantly improved PFS compared to Pem, Niv, PBC, with an HR and 95% CrI of 0.493 (0.331 to 0.706), 0.519 (0.317 to 0.805), and 0.397 (0.283 to 0.542), respectively. Pem significantly improved PFS compared to PBC, with an HR and 95% CrI of 0.814 (0.666 to 0.984). There was no significant difference between Niv+Ipi and Pem, between Niv+Ipi and Niv, between Pem and Niv, nor between Niv and PBC, with an HR and 95% CrI of 0.775 (0.563 to 1.042), 0.805 (0.637 to 1.005), 1.064 (0.712 to 1.530), 0.787 (0.558 to 1.079), respectively (Figure 4).

### 3.4. Secondary Efficacy Endpoint: Overall Survival in Patients with a PD-L1 of 1% or More

Niv+Ipi significantly improved OS compared to PBC, with an HR and 95% CrI of 0.793 (0.668 to 0.935), whereas Niv+Ipi was inferior in OS to Pem+PBC, with HR and 95% CrI of 1.465 (1.077 to 1.948). Pem+PBC significantly improved OS compared to Pem, Niv, and PBC, with an HR and 95% CrI of 0.681 (0.510 to 0.889), 0.631 (0.441 to 0.876), and 0.550 (0.428 to 0.696), respectively. Pem significantly improved PFS compared to PBC, with an HR and 95% CrI of 0.812 (0.708 to 0.927). There was no significant difference between Niv+Ipi and Pem, between Niv+Ipi and Niv, between Pem and Niv, nor between Niv and PBC, with an HR and 95% CrI of 0.982 (0.787 to 1.210), 0.903 (0.758 to 1.068), 0.931 (0.701 to 1.213), and 0.885 (0.691 to 1.116), respectively (Figure 5).

### 3.5. Secondary Efficacy Endpoint: Overall Survival in Patients with PD-L1 of 50% or More

Niv+Ipi significantly improved OS compared to PBC, with an HR and 95% CrI of 0.706 (0.547 to 0.896). Pem+PBC significantly improved OS compared to PBC, with an HR and 95% CrI of 0.513 (0.352 to 0.723). Pem significantly improved OS compared to PBC, with an HR and 95% CrI of 0.694 (0.560 to 0.849). There was no significant difference between Niv+Ipi and Pem+PBC, between Niv+Ipi and Pem, between Niv+Ipi and Niv, between Pem+PBC and Pem, between Pem+PBC and Niv, between Pem and Niv, nor between Niv and PBC, with an HR and 95% CrI of 1.423 (0.896 to 2.151), 1.029 (0.736 to 1.401), 0.877 (0.678 to 1.116), 0.748 (0.482 to 1.109), 0.648 (0.378 to 1.039), 0.875 (0.570 to 1.287), and 0.818 (0.567 to 1.143), respectively (Figure 6).

### 3.6. Primary Safety Endpoint: Incidence of G3–5AEs in the Overall Population

The incidence of G3–5AEs was significantly higher in Niv+Ipi compared to Pem or Niv, with an odds ratio (OR) and 95% CrI of 2.624 (1.813 to 3.674) and 1.973 (1.431 to 2.658), respectively. The incidence of G3–5AEs was significantly higher in Pem+PBC compared to Pem or Niv, with an OR and 95% CrI of 3.508 (2.408 to 4.953) and 2.680 (1.624 to 4.174), respectively. The incidence of G3–5AEs was significantly lower in Pem and Niv compared to PBC, with an OR and 95% CrI of 0.313 (0.240 to 0.400) and 0.419 (0.277 to 0.610), respectively. There was no significant difference between Niv+Ipi and Pem+PBC, between Niv+Ipi and PBC, between Pem+PBC and PBC, nor between Pem and Niv, with an OR and 95% CrI of 0.761 (0.525 to 1.068), 0.807 (0.626 to 1.022), 1.078 (0.829 to 1.379), and 0.777 (0.472 to 1.207), respectively (Figure 7).

### 3.7. Ranking Assessment

In addition, we ranked the efficacy and safety of the five treatment arms (Niv+Ipi, Pem+PBC, Pem, Niv, and PBC) by evaluating the surface under the cumulative ranking curve (SUCRA) values. With respect to efficacy, analysis was performed for each PFS and OS in a group with a PD-L1-positive rate of 1% or more and a group with a PD-L1-positive rate of greater than 50%. The data reported regarding the safety outcome (G3–5AEs) were insufficient to analyze PD-L1 positivity based on group. Therefore, safety was ranked based on the results for the overall population enrolled in each study. The results were represented by scatter plots showing the SUCRA values for efficacy (PFS and OS in groups with PD-L1-positive rates ≥1% and groups ≥50%) and safety (the G3–5AEs for all enrolled patients).

In the patient group with a PD-L1 of 1% or more, the SUCRA values for PFS revealed that Pem+PBC (SUCRA, 100%) ranked the best, Niv+Ipi (SUCRA, 74.6%) ranked second, PBC (SUCRA, 32.9%) ranked third, Niv (SUCRA, 32.2%) ranked fourth, and Pem (SUCRA, 10.4%) ranked fifth. In the patient group with PD-L1 of 50% or more, the SUCRA values for PFS revealed that Pem+PBC (SUCRA, 99.6%) ranked the best, Niv+Ipi (SUCRA, 73.5%) ranked second, Niv (SUCRA, 38.8%) ranked third, Pem (SUCRA, 36.0%) ranked fourth, and PBC (SUCRA, 2.1%) ranked fifth (Figure 8).

In the patient group with a PD-L1 of 1% or more, the SUCRA values for OS revealed that Pem+PBC (SUCRA, 99.7%) ranked the best, Niv+Ipi (SUCRA, 62.0%) ranked second, Pem (SUCRA, 53.3%) ranked third, Niv (SUCRA, 31.4%) ranked fourth, and PBC (SUCRA, 3.7%) ranked fifth. In the patient group with a PD-L1 of 50% or more, the SUCRA values for OS revealed that Pem+PBC (SUCRA, 95.6%) ranked the best, Niv+Ipi (SUCRA, 60.0%) ranked second, Pem (SUCRA, 59.5%) ranked third, Niv (SUCRA, 32.2%) ranked fourth, and PBC (SUCRA, 2.9%) ranked fifth (Figure 9).

The SUCRA values for the G3–5AEs revealed that Pem (SUCRA, 97.0%) ranked the best, Niv (SUCRA, 78.0%) ranked second, Niv+Ipi (SUCRA, 47.7%) ranked third, PBC (SUCRA, 18.4%) ranked fourth, and Pem+PBC (SUCRA, 8.9%) ranked fifth (Figure 8 and Figure 9).

### 3.8. Bias Assessment

We evaluated the risk of bias using the Cochrane risk of bias tool [20], and a low risk of bias was shown for all studies included in the present network meta-analysis. The risk of bias graph and risk of bias summary are shown in Appendix A.

### 3.9. Sensitivity Analysis

Of the studies included in this meta-analysis, KEYNOTE-407 had no restrictions on the EGFR gene mutation or ALK re-arrangement status. To assess the impact of excluding or including KEYNOTE-407 in the final conclusions, we performed a sensitivity analysis for efficacy outcome (PFS and OS). After excluding KEYNOTE-407, we found that the sensitivity analysis of almost all two-treatment comparisons for statistical significance tests delivered unchanged results, although the significant difference in PFS between Niv+Ipi and Pem+PBC in groups with a PD-L1 of 50% or more disappeared, and a significant difference in OS between Niv+Ipi and Niv in a PD-L1 of 50% or more appeared. Furthermore, there was little change in effect size and SUCRA values, and the results for the ranking assessment of the effectiveness of each treatment for primary and efficacy outcome were maintained in both the patient group with a PD-L1 of 1% or more and the patient group with a PD-L1 of 50% or more. In Appendix A section, we present the details of the results of sensitivity analysis. Based on these results, we believe that the inclusion/exclusion of KEYNOTE-407 did not affect our final conclusions.

## 4. Discussion

In the present meta-analysis, we compared the efficacy and safety of Niv+Ipi and existing regimens with immunotherapies with a PBC as a common comparator in patients with PD-L1-positive advanced NSCLC using the statistical method of a Bayesian NMA. The results showed that although PFS in the patient group with a PD-L1 expression of 1% or higher, which was the primary efficacy endpoint, was significantly better in the Pem+PBC than in the Niv+Ipi treatment group, the PFS was significantly better in Niv+Ipi than either Pem, Niv, or PBC. There were no significant differences in the G3–5AEs, which was the primary safety endpoint, nor among the Niv+Ipi-, Pem+PBC-, and PBC-treated patients; furthermore, the incidence of G3–5AEs was higher in Niv+Ipi than in either Niv or Pem. The SUCRA results revealed that Pem+PBC ranked highest in PFS, followed by Niv+Ipi, PBC, Niv, and PBC in the patient group with a PD-L1 expression of 1% or higher, and that Pem ranked highest in the G3–5AEs, followed by Niv, Niv+Ipi, PBC, and Pem+PBC in the overall population.

As secondary endpoints, we assessed PFS (in patients with a PD-L1 ≥50%), OS (in patients with a PD-L1 ≥1%), and OS (in patients with a PD-L1 ≥50%). In the analysis for each patient group, the effectiveness of Pem+PBS was ranked the best. Pem or Niv monotherapy was less effective than either Pem+PBS or Niv+Ipi, although the safety outcome was relatively good in the overall population. Niv+Ipi was ranked between Pem+PBS and Pem or Niv monotherapy in terms of efficacy in each patient group and safety in the overall population.

Previous studies have reported the comparative efficacy and safety between Pem+PBC and PBC or between Pem and PBC in PD-L1-positive NSCLC cases [21,35,36]. The results showed that Pem+PBC or Pem had a significantly better OS than PBC. It has also been reported that PFS was significantly better in Niv+Ipi than in the Niv or PBC groups [11]. Nonetheless, there has been no report comparing the efficacy and safety of Niv+Ipi and Pem+PBC or Pem.

This systematic review used an original approach as we compared and ranked the efficacy and safety of these treatment groups simultaneously. Our results will provide oncologists with clinical information on the efficacy and safety of Niv+Ipi, and also suggest the potential of Niv+Ipi as a new first-line treatment option for PD-L1-positive advanced NSCLC. Further, our results showed that Niv+Ipi has, regardless of the PD-L1 expression level, a better efficacy profile (PFS or OS) than Pem or Niv, which have already been approved as therapeutic options for PD-L1-positive advanced NSCLC. These results may be explained from the standpoint of molecular oncology.

Ipilimumab is a human IgG1-type monoclonal antibody against CTLA-4. Dendritic cells, in lymph nodes, repress activated T lymphocytes (effector T lymphocytes) through CD80/CD86 expression that bind to the T lymphocytes expressing CTLA-4 [37]. Ipilimumab activates and induces the proliferation of T lymphocytes by releasing T lymphocyte suppression by binding to the CTLA-4 on the T lymphocyte, thus blocking dendritic cell binding to CD80/CD86, which in turn re-promotes the cytotoxic activity of the T lymphocytes [38]. Niv is a monoclonal antibody belonging to the IgG4-type against PD-1 [39,40,41]. Niv releases the inhibitory state of the T lymphocytes by binding to the PD-1 molecule expressed by the T lymphocytes; which in turn blocks its binding to tumor cells expressing its ligand PD-L1, and thus reinstates the antitumor effect of the T lymphocytes [41,42] (Figure 10).

Thus, Ipi and Niv have different mechanisms of action, despite belonging to the same class of ICIs. Because Ipi and Niv complement each other in their ability to relieve immunosuppression, treatment with these agents together has a higher antitumor effect than using each agent alone [43,44,45]. The efficacy of Niv+Ipi therapy has been also confirmed for renal cell carcinoma and malignant melanoma, among others, and has already been applied clinically [46,47,48].

Conversely, Niv+Ipi was, in terms of the efficacy of PFS and OS, inferior to Pem+PBC, which is currently recommended as the first-line treatment for PD-L1-positive advanced NSCLC. We believe that identifying patient characteristics for which the outcome of Niv+Ipi treatment can be expected is an important future research area. Regarding safety outcomes, G3–5AEs showed no significant differences between Niv+Ipi and Pem+PBC, but SUCRA analysis indicated that Niv+Ipi was better tolerated than Pem+PBC. These results indicate that Niv+Ipi may be better tolerated compared to Pem+PBC and may be a useful treatment option in terms of safety.

This study has several limitations. First, because this study is a network meta-analysis of RCTs performed separately, the heterogeneity between the studies may not be negligible. The assessment of heterogeneity did not detect any significant heterogeneity, but may have influenced the final conclusion. Secondly, there were differences in the inclusion criteria of patients between the studies in this analysis. Of the four studies in this analysis, three included only EGFR and ALK-negative cases, but one (KEYNOTE-407) [21] included no restrictions on inclusion criteria for EGFR and ALK status. However, the latter study included only squamous cell carcinomas and may have had less impact on the final conclusions, given the low frequency of positive EGFR mutations and ALK gene translocations in squamous cell carcinomas. In fact, the results of our sensitivity analysis indicated that the final conclusion was unaffected by the inclusion or exclusion of KEYNOTE-407 [21]. Finally, the number of studies that were included was small. Between-study heterogeneity tends to influence the final conclusions in analyses from a small number of inclusion studies. Although we found no statistically significant between-study heterogeneity in the present study, it may have influenced the final conclusion.

## 5. Conclusions

In conclusion, we compared the efficacy and safety profiles of Niv+Ipi and existing immunotherapies in PD-L1-positive NSCLC, using PBC as a common comparator, via a Bayesian NMA. Our results revealed that, in terms of PFS, Niv+Ipi was inferior to Pem+PBC, and was superior to Pem, Niv, and PBC. No significant differences were observed in the frequency of G3–5AEs between Niv+Ipi and Pem+PBC, while the frequency of G3–5AEs was higher in the Niv+Ipi group than in the Pem- or Niv-treated groups. SUCRA results showed that Pem+PBC ranked highest in PFS, followed by Niv+Ipi, Niv, PBC, and Pem, and also revealed that Pem ranked highest in the G3–5AEs, followed by Niv, Niv+Ipi, PBC, and Pem+PBC. These results provide clinical information regarding the efficacy and safety of Niv+Ipi in PD-L1-positive advanced NSCLC, indicating the possibility of Niv+Ipi as a new therapeutic option as a first-line treatment for PD-L1-positive advanced NSCLC. Considering that this study is an NMA through direct and indirect comparison, verification by a direct head-to-head RCT is warranted to confirm the results obtained. Furthermore, identifying the characteristics of the patient populations where Niv+Ipi has particular benefit is an important future research topic.

## Figures and Tables

**Figure 1 cancers-12-01905-f001:**
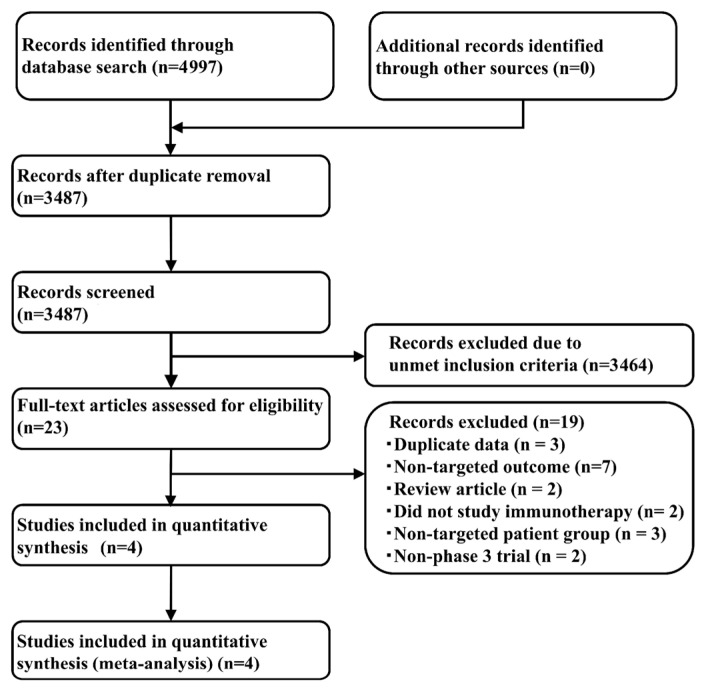
Flow diagram of the study selection process. RCT, randomized controlled trial.

**Figure 2 cancers-12-01905-f002:**
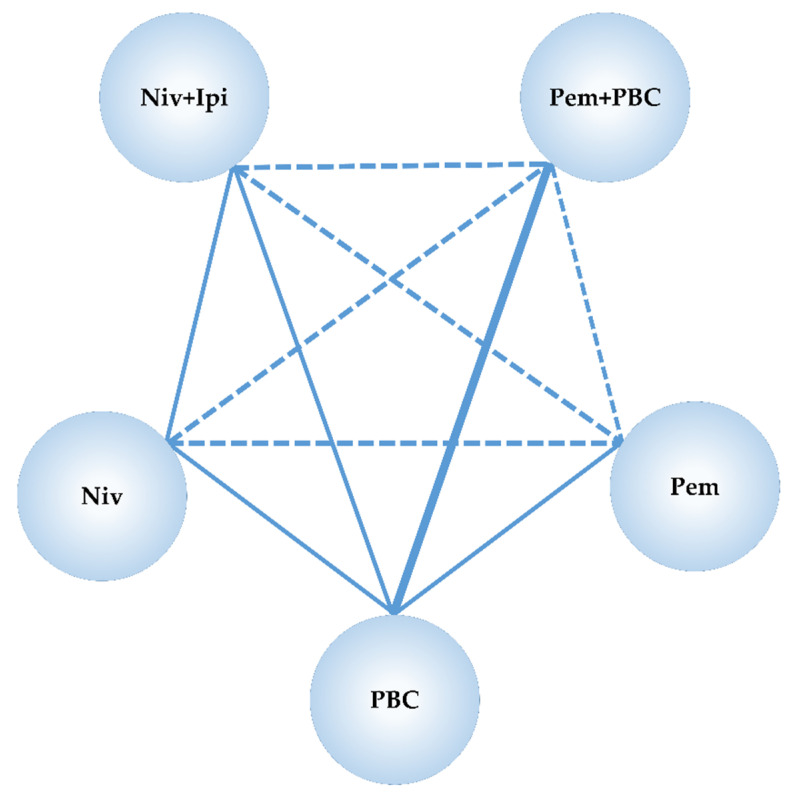
Network map of the network meta-analysis performed in this study. The randomized controlled trials (RCTs) included in this analysis are represented by solid lines, and the thickness of solid line indicates the number of included studies. The dashed line indicates a relationship where no RCT exists but an indirect comparison could be attempted. s.c., subcutaneous; RCT, randomized controlled trial; Niv, nivolumab; Ipi, ipilimumab; Pem, pembrolizumab; PBC, platinum-based chemotherapy.

**Figure 3 cancers-12-01905-f003:**
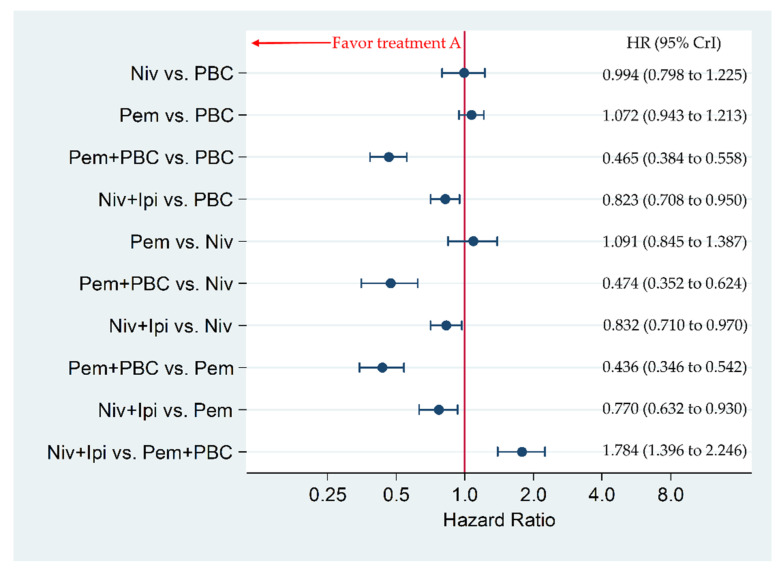
Comparative efficacy for progression-free survival (PFS) among five therapeutic regimens (Niv+Ipi, Pem+PBC, Pem, Niv, PBC) in patients with programmed cell death ligand 1 (PD-L1) expression levels of 1% or more. Comparisons were expressed as Treatment A versus Treatment B. Data are expressed in terms of hazard ratio (HR) and 95% credible intervals (CrIs); Niv, nivolumab; PBC, platinum-based chemotherapy; Pem, pembrolizumab; Ipi, ipilimumab.

**Figure 4 cancers-12-01905-f004:**
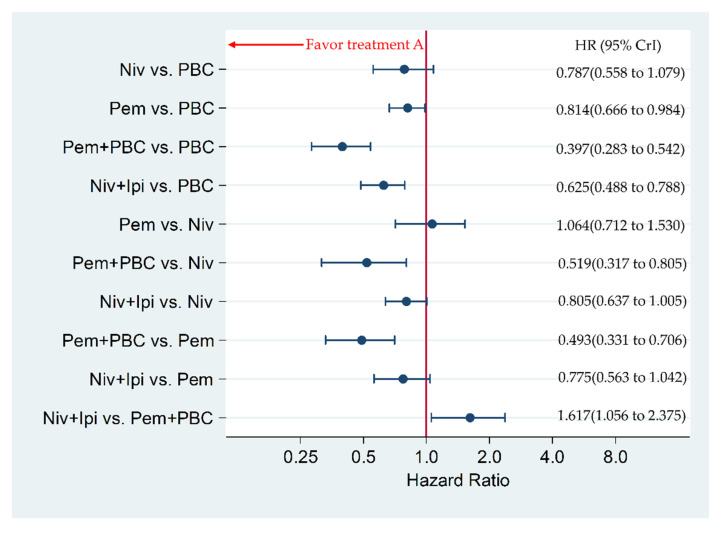
Comparative efficacy for progression-free survival (PFS) among the five therapeutic regimens (Niv+Ipi, Pem+PBC, Pem, Niv, and PBC) in patients with programmed cell death ligand 1 (PD-L1) expression levels of 50% or more. Comparisons were expressed as Treatment A versus Treatment B. Data are expressed in terms of hazard ratio (HR) and 95% credible intervals (CrIs); Niv, nivolumab; PBC, platinum-based chemotherapy; Pem, pembrolizumab; Ipi, ipilimumab.

**Figure 5 cancers-12-01905-f005:**
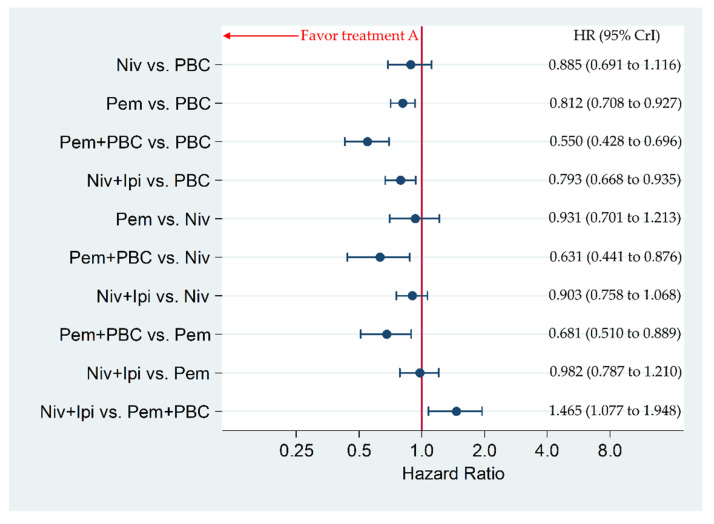
Comparative efficacy for overall survival (OS) among the five therapeutic regimens (Niv+Ipi, Pem+PBC, Pem, Niv, and PBC) in patients with programmed cell death ligand 1 (PD-L1) expression levels of 1% or more. Comparisons were expressed as Treatment A versus Treatment B. Data are expressed in terms of hazard ratio (HR) and 95% credible intervals (CrIs); Niv, nivolumab; PBC, platinum-based chemotherapy; Pem, pembrolizumab; Ipi, ipilimumab.

**Figure 6 cancers-12-01905-f006:**
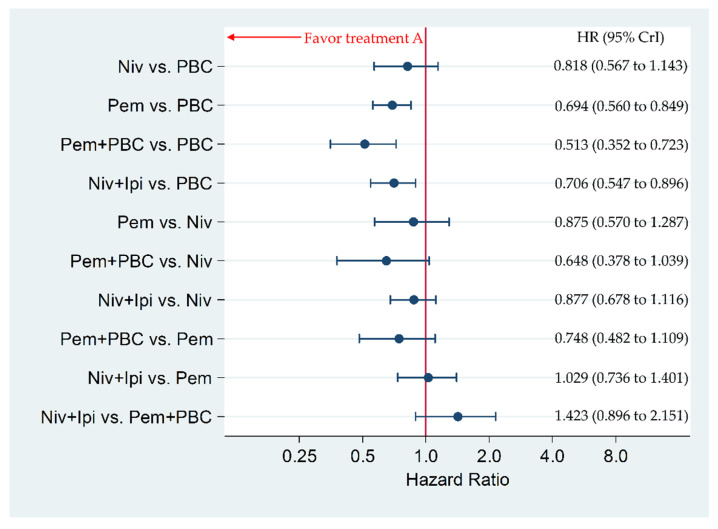
Comparative efficacy for overall survival (OS) among the five therapeutic regimens (Niv+Ipi, Pem+PBC, Pem, Niv, and PBC) in patients with programmed cell death ligand 1 (PD-L1) expression levels of 50% or more. Comparisons were expressed as Treatment A versus Treatment B. Data are expressed in terms of hazard ratio (HR) and 95% credible intervals (CrIs); Niv, nivolumab; PBC, platinum-based chemotherapy; Pem, pembrolizumab; Ipi, ipilimumab.

**Figure 7 cancers-12-01905-f007:**
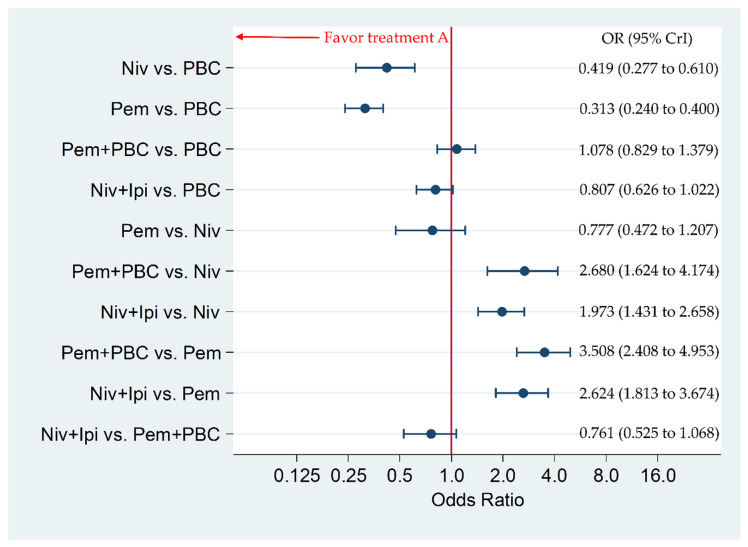
Comparative efficacy among the five therapeutic regimens (Niv+Ipi, Pem+PBC, Pem, Niv, and PBC) in terms of incidence of any Grade 3–5 drug-related adverse events (G3–5AEs) in the overall patient group. Comparisons were expressed as Treatment A versus Treatment B. Data are expressed in terms of odds ratio (OR) and 95% credible intervals (CrIs); Niv, nivolumab; PBC, platinum-based chemotherapy; Pem, pembrolizumab; Ipi, ipilimumab.

**Figure 8 cancers-12-01905-f008:**
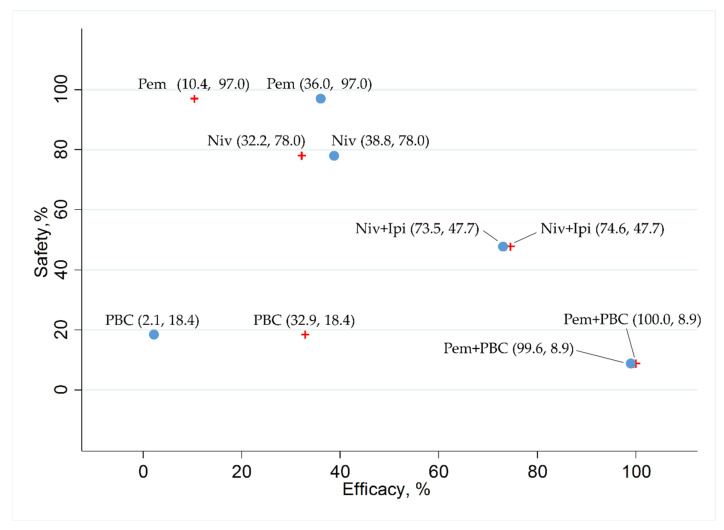
The surface under the cumulative ranking curve (SUCRA) of the efficacy in progression-free survival (PFS) in a group of patients with programmed cell death ligand 1 (PD-L1) expression levels of 1% or more (presented as red crosses), in a group of patients with PD-L1 expression levels of 50% or more (presented as blue circles), and safety in any Grade 3–5 drug-related adverse events (G3–5AEs) in the overall population of the five therapeutic regimens (Niv+Ipi, Pem+PBC, Pem, Niv, and PBC). Data are presented as (SUCRA in PFS and SUCRA in G3–5AEs) with each plot of the five therapeutic regimens. In terms of efficacy in patients with PD-L1 expression levels of 1% or more (presented as red crosses), Pem+PBC ranked the highest, followed by Niv+Ipi, PBC, Niv, and, finally, Pem. In terms of efficacy in patients with PD-L1 expression levels of 50% or more (presented as blue circles), Pem+PBC ranked the highest, followed by Niv+Ipi, Niv, Pem, and, finally, PBC. However, Pem ranked the highest in safety, followed by Niv, Niv+Ipi, PBC, and, finally, Pem+PBC. Niv, nivolumab; PBC, platinum-based chemotherapy; Pem, pembrolizumab; Ipi, ipilimumab.

**Figure 9 cancers-12-01905-f009:**
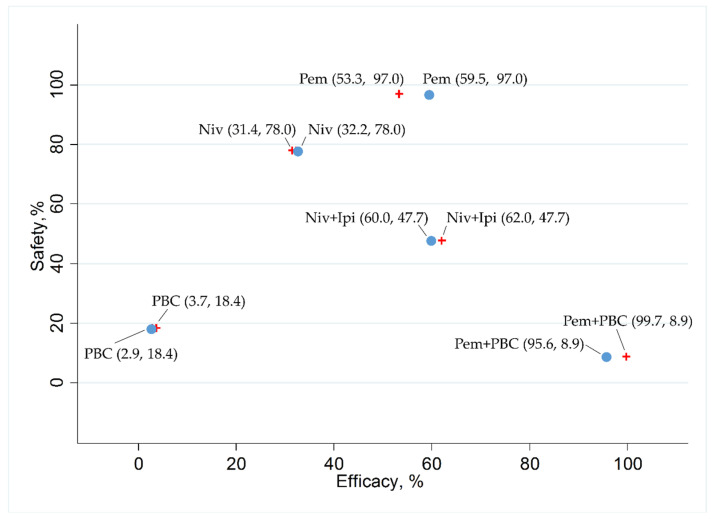
The surface under the cumulative ranking curve (SUCRA) of the efficacy in overall survival (OS) in a group of patients with programmed cell death ligand 1 (PD-L1) expression levels of 1% or more (presented as red crosses), in a group of patients with PD-L1 expression levels of 50% or more (presented as blue circles), and safety in any Grade 3–5 drug-related adverse events (G3–5AEs) in the overall population of the five therapeutic regimens (Niv+Ipi, Pem+PBC, Pem, Niv, and PBC). Data are presented as (SUCRA in OS and SUCRA in G3–5AEs) with each plot of the five therapeutic regimens. In terms of efficacy in patients with PD-L1 expression levels of 1% or more (presented as red crosses), Pem+PBC ranked the highest, followed by Niv+Ipi, Pem, Niv, and, finally, PBC. In terms of efficacy in patients with PD-L1 expression levels of 50% or more (presented as blue circles), Pem+PBC ranked the highest, followed by Niv+Ipi, Pem, Niv, and, finally, PBC. However, Pem ranked the highest in safety, followed by Niv, Niv+Ipi, PBC, and, finally, Pem+PBC. Niv, nivolumab; PBC, platinum-based chemotherapy; Pem, pembrolizumab; Ipi, ipilimumab.

**Figure 10 cancers-12-01905-f010:**
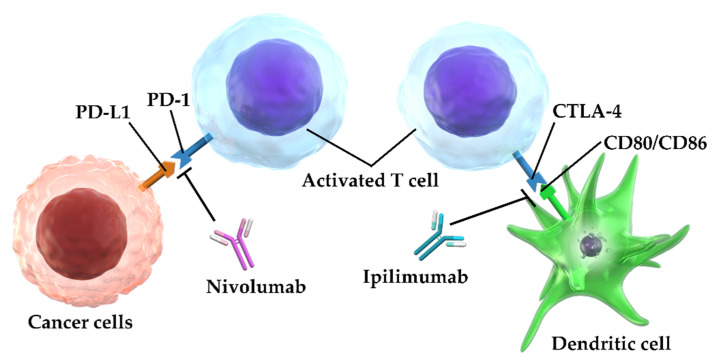
Ipilimumab releases T lymphocytes from suppression caused by CD80/CD86-expressing dendritic cells by competing for binding to CTLA-4 on T lymphocytes, and induces T lymphocyte activation and proliferation. Nivolumab releases T lymphocytes from suppression caused by PD-L1-expressing cancer cells by binding to the PD-1 molecule. These properties of ipilimumab and nivolumab contribute to the restoration of the antitumor effect of T lymphocytes; PD-1, programmed cell death-1; PD-L1, programmed cell death ligand 1; CTLA-4, cytotoxic T lymphocyte antigen-4.

**Table 1 cancers-12-01905-t001:** Key inclusion criteria of the included studies.

Study Names	Key Inclusion Criteria
KEYNOTE-189	18 years of age or older
	Metastatic non-squamous NSCLC with at least one measurable lesion Without a EGFR sensitive mutation or ALK fusion gene translocation
	No previous systemic anticancer treatment
	Performance status rank of 0 or 1
KEYNOTE-407	18 years of age or older
	Stage IV squamous cell carcinoma with at least one measurable lesion
	No previous systemic anticancer treatment
	Performance status rank of 0 or 1
KEYNOTE-042	18 years of age or older
	Locally advanced or metastatic NSCLC with at least one measurable lesion Without a EGFR sensitive mutation or ALK fusion gene translocation
	No previous systemic anticancer treatment
	Performance status rank of 0 or 1
	Having a PD-L1 expression proportion of 1% or greater.
CheckMata 227	18 years of age or older
	Squamous or non-squamous Stage IV or recurrent NSCLC Without a EGFR sensitive mutation or ALK fusion gene translocation
	No previous systemic anticancer treatment
	Performance status rank of 0 or 1

NSCLC, non-small cell lung cancer; EGFR, epithelial growth factor receptor; ALK, anaplastic lymphoma kinase; PD-L1, programmed cell death ligand 1.

**Table 2 cancers-12-01905-t002:** Characteristics of the included studies.

Study Name (Year of Publication)	Treatment Arms	N	Age—yrs. Median (range)	Female Sex No. (%)	ECOG PS No. (%)	Histologic Type No. (%)	PD-L1 Status
KEYNOTE-189 (2018)	Pembrolizumab 200 mg/body e3w plus platinum-based chemotherapy	410	65 (34–84)	156 (38.0)	PS0: 186 (45.4) PS1: 221 (53.9) PS2: 1 (0.2) Missing: 2 (0.5)	Adenocarcinoma 394 (96.1)	≥1% 260 (63.4)
						Other 16 (3.9)	≥50% 132 (32.2)
	Platinum-based chemotherapy	206	63.5 (34–84)	97 (47.1)	PS0: 80 (38.8)	Adenocarcinoma 198 (96.1)	≥1% 128 (62.1)
					PS1: 125 (60.7)	Other 8 (3.6)	≥50% 70 (34.0)
					Missing: 1 (0.5)		
		Total, 616					
KEYNOTE-407 (2018)	Pembrolizumab 200 mg/body e3w plus platinum-based chemotherapy	278	65 (29–87)	58 (20.9)	PS0: 73 (26.3)	Squamous: 272 (97.8)	≥1% 176 (63.3)
					PS1: 205 (73.7)	Adenosquamous: 6 (2.2)	≥50% 73 (26.3)
	Platinum-based chemotherapy	281	65 (36–88)	46 (16.4)	PS0: 90 (32.0)	Squamous: 274 (97.5)	≥1% 177 (63.0)
					PS1: 191 (68.0)	Adenosquamous: 7 (2.5)	≥50% 73 (26.0)
		Total, 559					
	Pembrolizumab 200 mg/body e3w	637	63.0 (57.0–69.0)	187 (29%)	PS0: 198 (31)	Squamous: 243 (38)	≥1% 637 (100)
					PS1: 439 (69)	Non-squamous: 394 (62)	≥50% 299 (46.9)
	Platinum-based chemotherapy	637	63.0 (57.0–69.0)	185 (29%)	PS0: 192 (30)	Squamous: 249 (39)	≥1% 637 (100.0)
					PS1: 445(70)	Non-squamous: 388(61)	≥50% 300 (47.1)
		Total, 1274					
CheckMate 227 (2019)	Nivolumab 3 mg/kg e2w plus ipilimumab 1 mg/kg e6w	583	64 (26–87)	190 (32.6)	PS0: 204 (35.0)	Squamous: 163 (28.0)	≥1% 396 (67.9)
					PS1: 377 (64.7)	Non-squamous: 419 (71.9)	≥50% 205 (35.2)
					Other or missing: 2 (0.3)	Missing data: 1 (0.2)	
	Nivolumab 240 mg/body e2w	396	64 (27–85)	124 (31.3)	PS0: 142 (35.9)	Squamous: 117 (29.5)	≥1% 396 (100.0)
					PS1: 252 (63.6)	Non-squamous: 279 (70.5)	≥50% 214 (54.0)
					Other or missing: 2 (0.5)	Missing data: 0 (0)	
	Platinum based chemotherapy	583	64 (29–87)	198 (34.0)	PS0: 191 (32.8)	Squamous: 162 (27.8)	≥1% 397 (68.1)
					PS1: 386 (66.2)	Non-squamous: 421 (72.2)	≥50% 192 (32.9)
					Other or missing: 6 (1.0)	Missing data: 0 (0)	
		Total, 1562					

The intention-to-treat (ITT) population contains all participants who were randomized, irrespective of whether an intervention was performed. N, number of patients; yrs., years; ECOG, Eastern Cooperative Oncology Group; PS, performance status; PD-L1, programmed cell death ligand 1; e3w, every three weeks; e2w, every two weeks, e6w, every 6 weeks.

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
