# Peer review of "Nivolumab plus Ipilimumab versus Existing Immunotherapies in Patients with PD-L1-Positive Advanced Non-Small Cell Lung Cancer: A Systematic Review and Network Meta-Analysis"

_cancers, 2020, doi:10.3390/cancers12071905_

Round 1

Reviewer 1 Report

The authors compared and ranked the efficacy and safety of Niv+Ipi and existing regimens with immunotherapies in PD-L1-positive advanced NSCLC patients. The analysis was valid and the results were interesting. However, I have following comments and suggestions.

  1. Too many figures. Please reorganize the figures. Some figures can be combined and some should be removed from the main text. For example, Figures 8 and 9 should be combined into one figure with two colors representing PD-L1 >1% and 50%. Figure 12 can be placed in supplementary file.
  2. As the authors pointed out, one of the study had no restriction on inclusion criteria for EGFR and ALK status. Would the findings remain if the study was excluded from the analysis? Please perform sensitivity analysis.

Author Response

Dear Reviewer 1

The authors compared and ranked the efficacy and safety of Niv+Ipi and existing regimens with immunotherapies in PD-L1-positive advanced NSCLC patients. The analysis was valid and the results were interesting. However, I have following comments and suggestions. 

Thank you for giving us the opportunity to submit to Cancers a revised draft of our manuscript titled “Nivolumab plus ipilimumab versus existing immunotherapies in patients with PD-L1-positive advanced non-small cell lung cancer: A systematic review and network meta-analysis”. We sincerely appreciate the time and effort that you and the reviewers have dedicated to providing your valuable feedback on our manuscript. We are grateful to the reviewers for their insightful comments on our paper. We have incorporated changes to reflect most of the suggestions provided by the reviewers. The changes within the manuscript are highlighted in red.

We present a point-by-point response to the reviewers’ comments and concerns.

Comment1: Too many figures. Please reorganize the figures. Some figures can be combined and some should be removed from the main text. For example, Figures 8 and 9 should be combined into one figure with two colors representing PD-L1 >1% and 50%. Figure 12 can be placed in supplementary file.

Response 1: We agree. Following the reviewer's comments, we have combined former figures 8 and 9, and renamed them Figure 8. In addition, we have combined former figures 10 and 11, and renamed them Figure 9. Accordingly, we have revised the figure legend for each new figure (lines 327–339, 349–360). We have moved former Figure 12 to Supplementary Materials and this move is reflected in the text (line 369).

Comment2: As the authors pointed out, one of the study had no restriction on inclusion criteria for EGFR and ALK status. Would the findings remain if the study was excluded from the analysis? Please perform sensitivity analysis.

Response 2: We agree. Thank you for raising this very important point. Following the reviewer's comments, we performed a sensitivity analysis excluding KEYNOTE-407, a study with no restrictions on EGFR mutations or ALK rearrangement status in its inclusion criteria. The sensitivity analysis performed after excluding KEYNOTE-407 revealed the same results for almost all two-treatment comparisons for statistical significance tests, although significant differences in PFS between Niv+Ipi and Pem+PBC in groups with PD -L1 of 50% or more disappeared; in addition, we found a significant difference in OS between Niv+Ipi and Niv in PD-L1 >50% or more. Furthermore, there was little change in effect size and SUCRA values, and the results for the ranking assessment of the effectiveness of each treatment for primary and efficacy outcome were maintained in both the patient group with PD-L1 of 1% or more and the patient group with PD-L1 of 50% or more. In Table S1 and Table S2 in Supplementary Materials, we present the details of the results of the sensitivity analysis. Based on these results, we believe that the inclusion/exclusion of KEYNOTE-407 did not affect our final conclusions (lines 371–385)。In addition, when discussing limitations in the Discussion section, we added a description regarding the sensitivity analysis (lines 457–459).

We are confident that our revised manuscript will be suitable for publication in Cancers and look forward to receiving your editorial decision.

Thank you for your consideration.

Sincerely,

Koichi Ando

Department of Medicine, Division of Respiratory Medicine and Allergology, Showa University School of Medicine

1-5-8 Hatanodai, Shinagawa-ku, Tokyo, 142-8666, Japan

Tel: +81-3-3784-8532

Fax: +81-3-3784-8742

Email: koichi-a@med.showa-u.ac.jp

Reviewer 2 Report

  • Overall this is a well written and well executed comparison of efficacy and safety profiles of Niv+Ipi and existing immunotherapies in PD-L1-positive NSCLC using PBC as a common comparator by Bayesian NMA.
  • This provides novel data comparing directly and indirectly the efficacy and safety of Niv+Ipi vs Pem+PBC or single agent Pem, Niv, or PBC
  • The stats and conclusions seem sound, although clearly RCT data preferable, the suggestion is that Niv+Ipi as a new first-line treatment option for PD-L1-positive advanced NSCLC is reasonable& further that Niv+Ipi has, beyond PD-L1 expression levels, a better efficacy profile.
  • Conclusions are interesting and thus will attract a wide readership in particular-  oncology/ resp clinicians.
  • Limitations of the study are recognised and discussed ie heterogeneity between studies influencing conclusion and differing inclusion criteria between studies.

Author Response

Dear Reviewer 2

Comment: Overall this is a well written and well executed comparison of efficacy and safety profiles of Niv+Ipi and existing immunotherapies in PD-L1-positive NSCLC using PBC as a common comparator by Bayesian NMA.

This provides novel data comparing directly and indirectly the efficacy and safety of Niv+Ipi vs Pem+PBC or single agent Pem, Niv, or PBC

The stats and conclusions seem sound, although clearly RCT data preferable, the suggestion is that Niv+Ipi as a new first-line treatment option for PD-L1-positive advanced NSCLC is reasonable& further that Niv+Ipi has, beyond PD-L1 expression levels, a better efficacy profile.

Conclusions are interesting and thus will attract a wide readership in particular-  oncology/ resp clinicians.

Limitations of the study are recognised and discussed ie heterogeneity between studies influencing conclusion and differing inclusion criteria between studies.

Response: Thank you for your insightful comments regarding our manuscript submitted to Cancers and titled “Nivolumab plus ipilimumab versus existing immunotherapies in patients with PD-L1-positive advanced non-small cell lung cancer: A systematic review and network meta-analysis”. We sincerely appreciate the time and effort that you and the reviewers have dedicated to providing your valuable feedback on our manuscript.

We are confident that our revised manuscript will be suitable for publication in Cancers and look forward to receiving your editorial decision.

Thank you for your consideration.

Sincerely,

Koichi Ando

Department of Medicine, Division of Respiratory Medicine and Allergology, Showa University School of Medicine

1-5-8 Hatanodai, Shinagawa-ku, Tokyo, 142-8666, Japan

Tel: +81-3-3784-8532

Fax: +81-3-3784-8742

Email: koichi-a@med.showa-u.ac.jp

Reviewer 3 Report

This systematic review is well written and analyzed. The strategy to use a network metanalysis allows to overcome the limit of few studies analyzed. I have doubts about the solidity of the results given the small amount of data and studies. The methods section should precede the results section. English language and style are fine, minor spell check required.

Author Response

Dear Reviewer 3

This systematic review is well written and analyzed. The strategy to use a network metanalysis allows to overcome the limit of few studies analyzed. English language and style are fine, minor spell check required.

Thank you for giving us the opportunity to submit to Cancers a revised draft of our manuscript titled “Nivolumab plus ipilimumab versus existing immunotherapies in patients with PD-L1-positive advanced non-small cell lung cancer: A systematic review and network meta-analysis”. We sincerely appreciate the time and effort that you and the reviewers have dedicated to providing your valuable feedback on our manuscript. We are grateful to the reviewers for their insightful comments, and have been able to incorporate changes to reflect most of their suggestions. We have highlighted in red the changes within the manuscript.

Here is a point-by-point response to the reviewers’ comments and concerns.

Comment1: I have doubts about the solidity of the results given the small amount of data and studies.

Response1: We agree. Thank you for raising a very important point. We have identified the small number of included studies as one of the limitations in this study. We have modified the discussion section to include the following: “Finally, the number of studies that were included was small. In analyses from a small number of inclusion studies, between-study heterogeneity tends to influence the final conclusions. Although we found no statistically significant between-study heterogeneity in the present study, it may have influenced the final conclusion”(lines 459–462).

Comment2: The methods section should precede the results section.

Response2: We agree. Following reviewer’s comment, we placed the method section before the result section, and revised section numbers and reference numbers accordingly.

We are confident that our revised manuscript will be suitable for publication in Cancers and look forward to receiving your editorial decision.

Thank you for your consideration.

Sincerely,

Koichi Ando

Department of Medicine, Division of Respiratory Medicine and Allergology, Showa University School of Medicine

1-5-8 Hatanodai, Shinagawa-ku, Tokyo, 142-8666, Japan

Tel: +81-3-3784-8532

Fax: +81-3-3784-8742

Email: koichi-a@med.showa-u.ac.jp